# Using a physics-informed neural network and fault zone acoustic monitoring to predict lab earthquakes

Prabhav Borate[1], Jacques Rivière[1], Chris Marone[2,3], Ankur Mali [4], Daniel Kifer[5] & Parisa Shokouhi [1] ✉

Predicting failure in solids has broad applications including earthquake prediction which remains an unattainable goal. However, recent machine learning work shows that laboratory earthquakes can be predicted using micro-failure events and temporal evolution of fault zone elastic properties. Remarkably, these results come from purely data-driven models trained with large datasets. Such data are equivalent to centuries of fault motion rendering application to tectonic faulting unclear. In addition, the underlying physics of such predictions is poorly understood. Here, we address scalability using a novel Physics-Informed Neural Network (PINN). Our model encodes fault physics in the deep learning loss function using time-lapse ultrasonic data. PINN models outperform data-driven models and significantly improve transfer learning for small training datasets and conditions outside those used in training. Our work suggests that PINN offers a promising path for machine learning-based failure prediction and, ultimately for improving our understanding of earthquake physics and prediction.

Prediction of catastrophic failure remains a critical albeit challenging endeavor across disciplines, from the nondestructive evaluation and structural health monitoring of industrial components[1,2] and civil infrastructure[3–5] to geophysics. In the latter domain, decades of research have greatly improved our understanding of earthquake physics, however, it is not yet possible to make sufficiently accurate predictions of when/where destructive earthquakes will occur, and we can currently only rely on seismic hazard maps that tell us about the likelihood for a magnitude-$x$ earthquake to strike a particular region in the next $y$ years[6,7]. In the last decades, anthropogenic earthquakes near geothermal reservoirs or due to wastewater injection have also threatened communities in regions of historically low seismicity[8,9], sometimes leading to early termination of innovative, costly projects[10–13]. Improving our ability to assess seismic risk—or in the long-term forecast earthquakes—would have a strong societal impact, saving lives, reducing economic disaster, as well as strengthening our ability to produce geothermal energy and non-conventional oil & gas by mitigating seismic risk near production sites.

Numerous laboratory studies have shown that the onset of failure is associated with bursts of acoustic emission (AE) events taking place during crack initiation and growth, and the number and amplitude of AE events generally increase as the sample approaches failure[14–24]. Recent friction studies on laboratory faults have shown that machine learning (ML) algorithms can actually predict the timing and magnitude of lab quakes using AE data[15,16,25–32]. It is remarkable that solely using acoustic emission data radiating from the faults as an input, the fault strength can be accurately predicted throughout the laboratory seismic cycle[25,27]. Past work[33–35] has shown that the vast majority of events radiate from the fault plane, therefore carrying information

[1]Department of Engineering Science and Mechanics, The Pennsylvania State University, University Park, PA 16802, USA. [2]Dipartimento di Scienze della Terra, La Sapienza Università di Roma, Roma, Italy. [3]Department of Geosciences, The Pennsylvania State University, University Park, PA 16802, USA. [4]Department of Computer Science and Engineering, University of South Florida, Tampa, FL 33620, USA. [5]Department of Computer Science and Engineering, The Pennsylvania State University, University Park, PA 16802, USA. ✉e-mail: pxs990@psu.edu

about the fault state. And as the elastic waves radiate/scatter through the host granite blocks, they also provide information about the stress state of the host rock. It is also remarkable that predictions work in the early stage of the seismic cycle when the acoustic signal often looks like noise, either because it lacks a clear P-wave, such as expected for friction/fracture events, or because it represents something like tectonic tremor involving the sum of many small or low frequency events that overlap in time and cannot be distinguished as separate events[36]. These studies show that the variance and kurtosis of AE data are the most predictive features among the ~100 features considered by ML models[15,27]. Other studies using active-source ultrasonic measurements have shown that laboratory quakes are preceded by reliable precursory signals, such as systematic changes in elastic wave velocity and amplitude[37–41]. Most recent studies have shown that laboratory quakes can also be predicted from active-source measurements using machine learning approaches[42,43], despite unfavorable conditions such as the occurrence of irregular seismic cycles. Again, it is remarkable that for some of the deep learning approaches used (like Long Short-Term Memory or LSTM), the $R^2$ values reach ~0.94 for shear stress prediction.

While greatly striking, laboratory quake predictions are enabled by the large amount of training data available. Scaled up to geological times, it would be equivalent to decades/centuries of data in nature. Also, such models might perform well for one particular dataset but fail to provide accurate predictions for another, slightly different dataset. This challenge can be overcome using various strategies such as meta-learning[44,45] and continual lifelong learning[46]. An alternative promising approach is Physics-Informed Neural Network (PINN) modeling[47] with the goal of improving predictions, model transferability and generalizability while reducing the amount of required training data. Physics-informed learning integrates pure data and physical laws to train the models. The PINN models can be implemented by introducing observational, inductive, or learning bias[47]. In case of observational bias, sufficient data covering the input and output domain of a learning model serve as a form of physics-based constraint that is embedded in the ML model[48]. The main challenge is the cost of data acquisition to generate a large volume of data which may involve complex and large-scale experiments or computational models. The inductive biases approach focuses on developing specialized neural network architectures that implicitly incorporate physics[49]. The effectiveness of these models is currently limited to simple physics and their extension to complex laws is still challenging and difficult to encode in the network architecture. Finally, the PINN can be implemented using the learning bias approach in which appropriate physics constraints are added to the cost function to penalize predictions inconsistent with the underlying physics[50,51]. This approach is widely used as the flexibility of adding penalty constraints allows the inclusion of many domain-specific physical principles into the model.

Here, we present a learning bias-based PINN framework that is tasked with predicting shear stress and fault slip rate history given information on fault zone elastic wave speed and transmitted amplitude. The PINN framework incorporates two physics constraints, one that relates the elastic coupling of a fault with the surrounding host rock[52], and another that relates fault stiffness to the ultrasonic transmission coefficient[53,54]. We systematically vary the amount of training data and find that, as training data becomes scarce, the PINN models outperform the purely data-driven models by roughly 10–15%. The PINN models are also more effective than purely data-driven models when tasked to predict laboratory quakes from a differing dataset (transfer learning).

## Results and discussion

A brief description of the experimental procedure and data is given below. The performance of data-driven, PINN and transfer learning models for different training data sizes is then presented and discussed.

### Friction experiment & ultrasonic monitoring

The friction experiments are carried out in a double direct shear (DDS) loading configuration, where three blocks of Westerly granite are loaded to create two faults, each of area $5 \times 5$ cm$^2$ (inset, Fig. 1a). Prior to loading, the block surfaces are dusted with fine quartz powder ($< \approx 200$ μm layer thickness). The vertical and horizontal pistons are used to apply shear ($\tau$) and normal ($\sigma$) stresses, respectively. Two direct current differential transformers (DCDT) are used to measure shear and normal displacements of the hydraulic pistons, while two strain-gauge load cells measure shear and normal stresses. An additional on-board DCDT attached to the central granite block and referenced to the base of the sample assembly is used to measure slip rate ($V$). To commence the experiment, normal stress $\sigma = 10$ MPa is applied to the sample and held constant via servo-control. Next, the central block is pushed downward at a constant rate of $V_l = 8.9$ μm/s. As the block is pushed down, the shear stress increases until the fault becomes unstable, the shearing block rapidly displaces (slips) and a sharp drop in the shear stress is recorded (lab earthquake). After the drop, a new stick-slip cycle begins; the fault gets locked again (sticks) and shear stress continues to increase. In order to develop robust machine learning models using these data, it is desired to create a frictional regime that produces irregular stick-slips. To that end, an acrylic spring is placed in series between the vertical piston and the central block to reduce the overall stiffness of the loading apparatus ($K$), and experiments are conducted close to the stability boundary producing both regular and irregular stick-slip cycles[55]. In this study, we use data collected in two separate experiments[41] namely, p5270 and p5271. The only difference between the two is the size of the acrylic spring used: 25 cm$^2$ for experiment p5270 and 20.25 cm$^2$ for p5271 resulting in different recurrence intervals and magnitudes of the lab quakes as shown in Supplementary Figs. S1 and S2. In p5270, the cycles become larger much earlier in the experiment. On the contrary, in p5271 they become larger later in the experiment. Throughout the experiment, all the stresses and displacements (including shear stress and slip rate) are recorded at a rate of 10 kHz. Additional details about these experiments can be found in refs. 40,41.

The friction experiment is coupled with active source ultrasonic monitoring. A pair of p-wave ultrasonic transducers are used to regularly probe the faults throughout the experiment. The two identical piezoelectric disks, used as transmitter and receiver, are 12.7 mm in diameter, 4 mm thick (corresponding to a center frequency of 500 kHz) made of material 850 from American Piezo Ceramics (APC International). The piezoelectric transducers are epoxy-glued at the bottom of blind holes inside steel platens that hold the DDS assembly. The transmitter $T$ imparts a series of half-sine pulses at 500 kHz every 1 ms throughout the experiment (Fig. 1b). The response is recorded by the receiving transducer $R$ at a sampling rate of 25 MHz (Fig. 1c). For experiments p5270 and p5271, the recorded ultrasonic data consist of 132,399 and 75,999 signals recorded during 387 and 237 stick-slip cycles, respectively.

### Ultrasonic feature extraction

Physics-based features, namely wave speed ($v_i$) and spectral amplitude ($A_i$) at time $t_i$, are extracted from each ultrasonic signal (waveform)[42]. Figure 2 illustrates the feature extraction process. To calculate the evolution of wave speed during frictional sliding, we first extract the time delay $\Delta t$ by cross-correlating each waveform $S_i$ with a reference waveform $S_0$. The reference waveform is chosen past the peak friction just before the fault starts its transition from stable sliding to unstable seismic cycles (thin vertical dashed line at time = 2065 s in Fig. 1a). The shape of the recorded waveforms $S_i$ changes little throughout the experiment such that the cross-correlation coefficient remains always

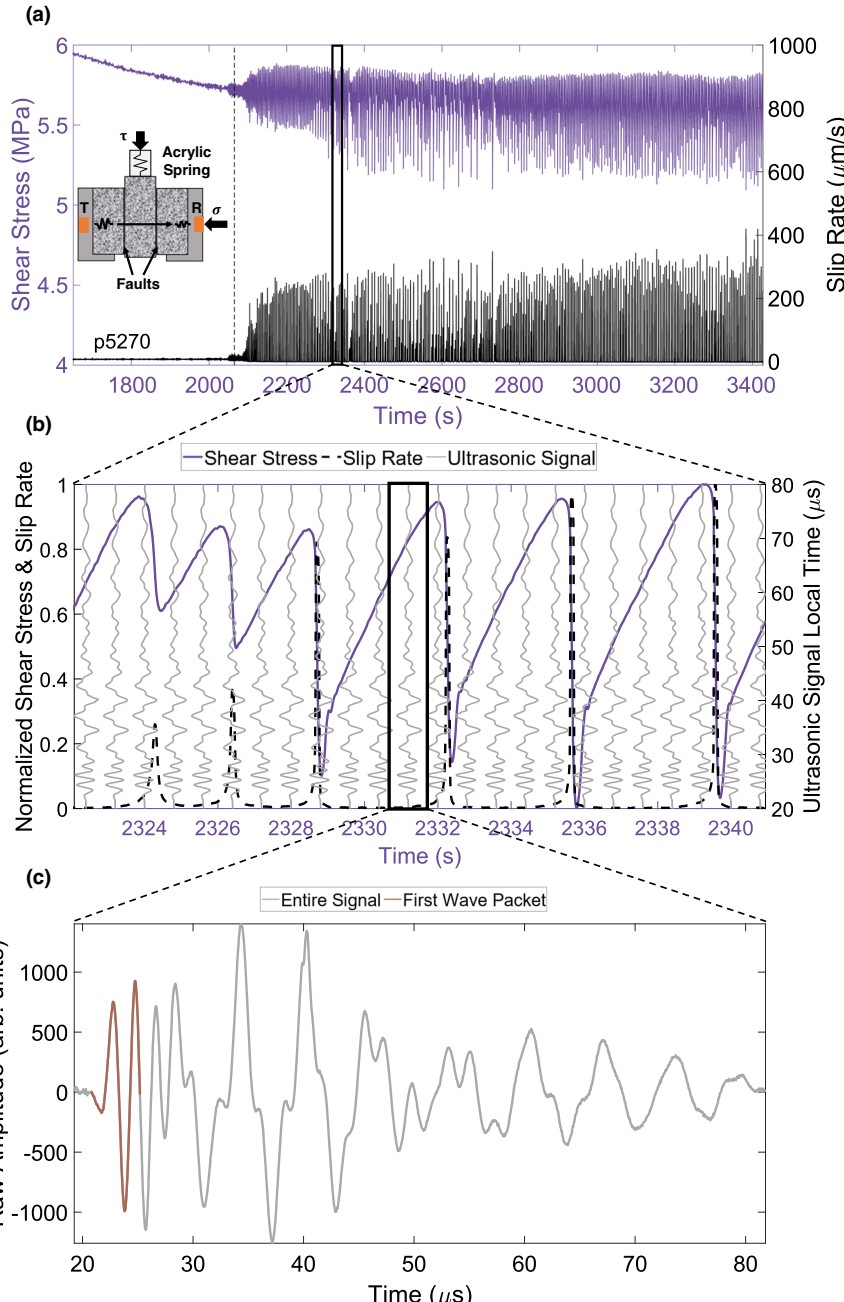

**Fig. 1 | Friction experiment coupled with ultrasonic monitoring: schematic setup and typical data. a** Temporal evolution of shear stress and slip rate in experiment p5270. The inset shows a schematic of the DDS setup with two ultrasonic transducers (transmitter T and receiver R) probing the fault. The thin vertical dashed line corresponds to the time at which the reference ultrasonic waveform is chosen (see text for more details). **b** Schematic representation of active-source ultrasonic monitoring during the experiment. The ultrasonic waveforms are recorded every millisecond throughout the stick-slip cycles. Only a small subset of the waveforms is shown for readability. **c** An example of a recorded ultrasonic signal. Input features to the machine learning models are extracted from the initial portion of the ultrasonic signals (highlighted in brown).

greater than 0.97. The cross-correlation is calculated within a finite window of size $T$ extending from $t_i + w_1$ to $t_i + w_2$ as shown in Fig. 2a. The peak of the cross-correlation between the two waveforms is refined by fitting a parabola passing through the peak and two adjacent points. For each signal $S_i$, the total travel time or time-of-flight $TOF_i$ is obtained by adding the hand-picked arrival time of the reference waveform ($TOF_0$) to the estimated time delay ($TOF_i = TOF_0 + \Delta t_i$). Finally, the wave speed is calculated by dividing the sample thickness by the travel time ($v_i = h_i / TOF_i$), where $h_i = h_0 + \delta_h$ and $h_0$ is the thickness measured at the beginning of the experiment just after applying the normal stress and $\delta_h$ is the thickness change measured continuously during the experiment.

The second feature, spectral amplitude $A_i$, is calculated as the amplitude of the Fourier transform of each signal $S_i$ within a finite window of size $T$ extending from $t_i + w_1$ to $t_i + w_2$ at a frequency of 400 kHz close to the center frequency of transmitted waves as shown in Fig. 2b. To reduce noise, both feature histories are low-pass filtered using a 10-point backward-looking moving average.

**Performance of data-driven, PINN and transfer learning models**

We compare the performance of PINN vs data-driven models as well as transfer-learned vs stand-alone models for different training-validation-test splits. The data-driven models are developed

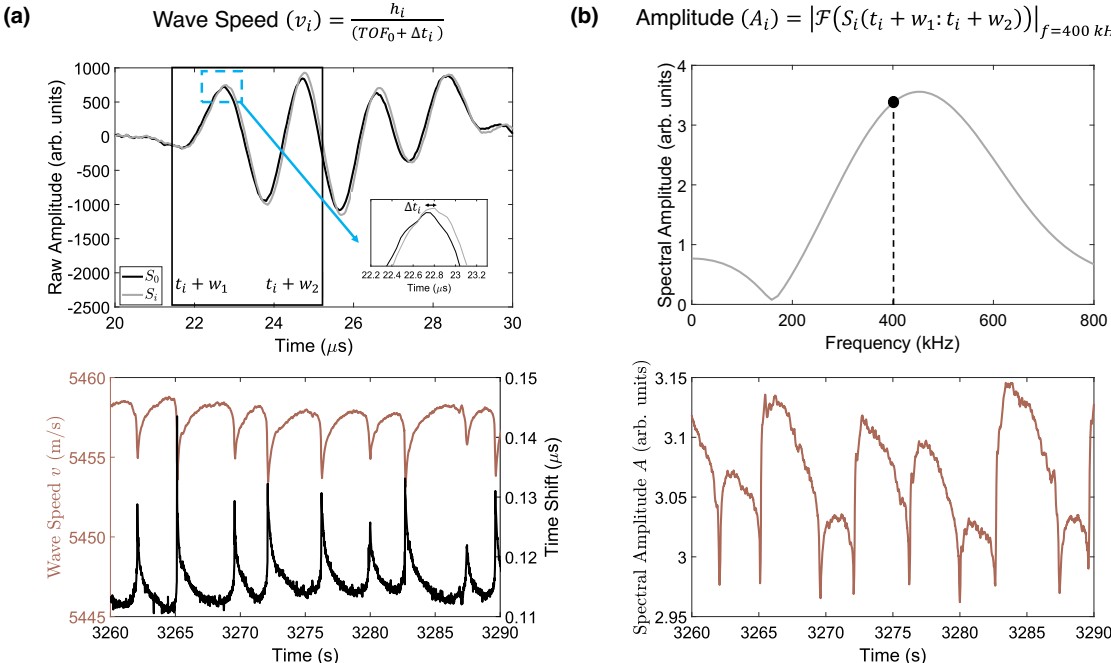

**Fig. 2 | Details of the feature extraction procedure. a** Shows the reference waveform ($S_0$) and a typical waveform during shearing ($S_i$). The inset emphasizes the time delay between the two signals $\Delta t_i$ calculated by cross-correlating the two signals. The box marks the extent of the cross-correlation window from $t_i + w_1$ to $t_i + w_2$ with $w_1 = 20.76$ μs and $w_2 = 25.16$ μs. The bottom plot shows a sample of wave speed and time shift evolution for several lab seismic cycles over a period of 30 s.

**b** Illustrates the spectral amplitude calculation from the Fourier spectrum of the windowed signal. The plot at the bottom shows an exemplary evolution of spectral amplitude. Note that wave speed and amplitude vary systematically with shear stress, but have a complex nonlinear relationship with shear stress as given in ref. 42.

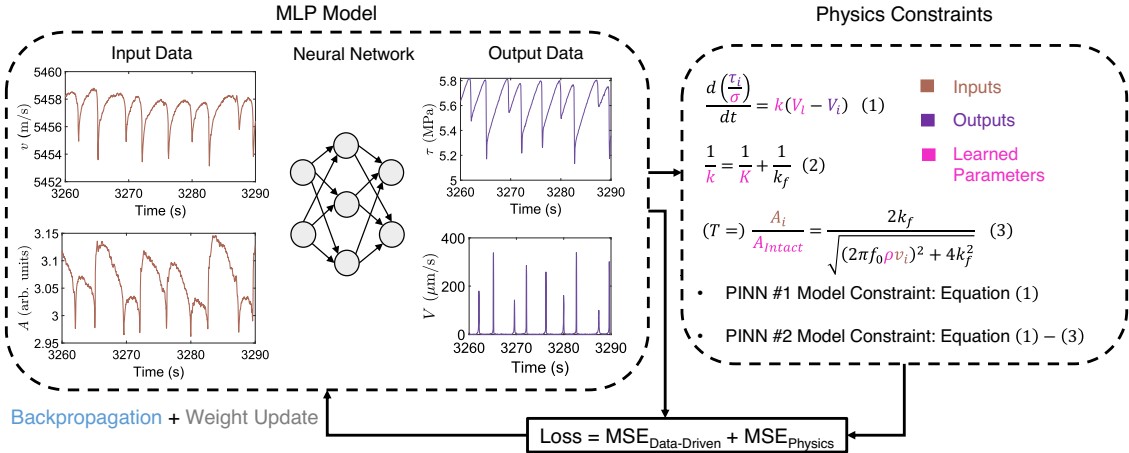

**Fig. 3 | Schematic representation of the PINN framework and Multilayer Perceptron (MLP) structure used for shear stress ($\tau_i$) and slip rate ($V_i$) prediction.** The loss function includes both data-driven ($MSE_{Data–Driven}$) and physics ($MSE_{Physics}$) losses. We explore two PINN models: PINN #1 and PINN #2.

using the Multilayer perceptron (MLP) neural network with a back-propagation algorithm[56] and Adam optimizer[57] to perform the regression task. The PINN models are built upon the data-driven models with the loss function modified to include the physics-based constraints as shown schematically in Fig. 3. Two different PINN models (PINN #1 and PINN #2) are considered. The PINN #1 model is constrained by the elastic coupling relation of a fault with the surrounding host rock (see Eq. (1)). The PINN #2 model is constrained also by the coupling (Eq. (1)) of fault stiffness to the ultrasonic transmission coefficient relations (Eqs. (2) and (3)). The data-driven, PINN #1, and PINN #2 models share the same MLP framework (hidden layers, units, batch size, optimizer, and learning rate) across different data splits to allow one on one comparison. Transfer learning (TL) models for p5271 experiment initialize with the pre-trained

p5270 model weights as schematically illustrated in Fig. 4. The performance of the TL models is compared with the standalone data-driven models for p5271 experiment. A detailed description of the data-driven models as well as the PINN and transfer learning frameworks including data selection and normalization is provided in "Methods".

All the reported models are developed with Google Colab using GPU acceleration with 16 GB memory. Each model is re-run entirely with three random seed instances to obtain an estimate of the model variance. The average $R^2$ score with standard error as well as the average root mean square error (RMSE) values and training times are reported for all the models.

Data-driven multi-output MLP models developed using the p5270 experimental dataset serve as a reference for later comparisons with

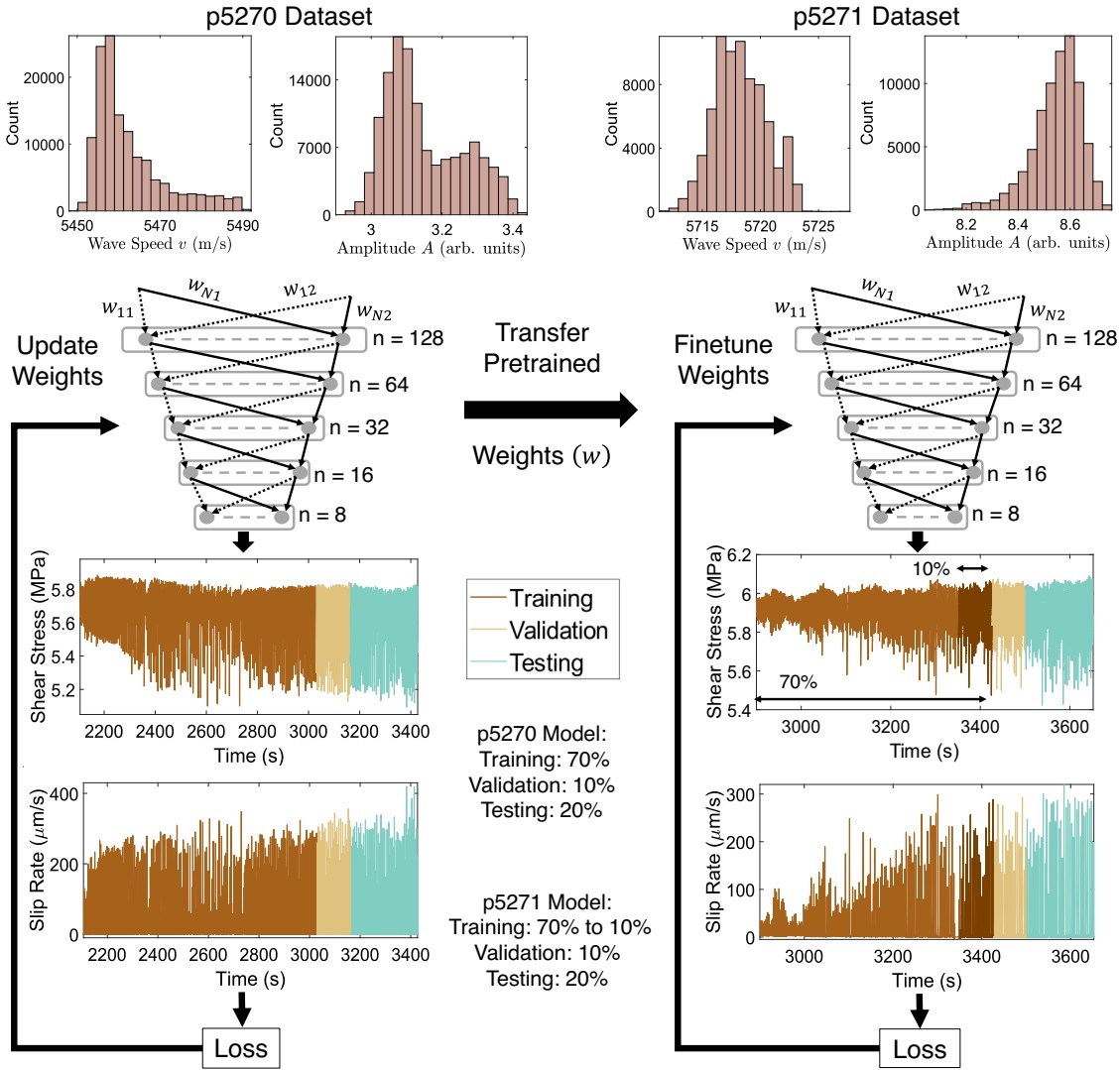

**Fig. 4 | Proposed transfer learning approach.** Best performing data-driven & PINN model's weights developed using the p5270 dataset are transferred and fine-tuned to build TL data-driven & TL PINN models for predicting shear stress and slip rate in the p5271 dataset.

PINN models. The reference model is developed with the training set size varying from 70% down to 5%. The $R^2$ scores of the reference model for predicting the shear stress and slip rate using training, validation, and testing sets as a function of varying training set sizes are plotted in Fig. 5 (grey bars). We observe that models trained with more than 20% of training data result in test $R^2$ scores greater than 0.9 for shear stress prediction. For slip rate prediction, $R^2$ scores are between 0.75 to 0.87. In all the cases considered, shear stress is predicted more accurately than the slip rate. Further reduction in the amount of training data (10% and less) results in a considerably lower test $R^2$ score, although the training $R^2$ scores remain reasonably high, a possible indication of overfitting. In other words, although the models with reduced data fit the training data well, they poorly fit the nonlinear relationship between the ultrasonic features (input) and shear failure variables (target) in validation and test sets. These results serve as a baseline performance to evaluate the performance of PINN models.

Comparisons of PINN #1 and PINN #2 models with the reference data-driven model for varying training set sizes are shown in Fig. 5 (brown and yellow bars, respectively). Similar to reference models, both PINN models show test $R^2$ scores greater than 0.9 when training data are 20% or more. Importantly, for all the considered splits, the PINN models perform equally well or better than their data-driven counterparts. Furthermore, the performance improvement is most evident when the training data are scarce (20% and less), especially in slip rate prediction. The PINN models trained on small sets (20% and less) also show less $R^2$ score variance, which suggests that the models are stable and result in a small variation in the prediction of the target data with changes in the model initialization set by random seeds. Table 1 compares the RMSE values for the Reference data-driven, PINN #1, and PINN #2 models. Note that the RMSE values (calculated using normalized data) are consistently larger for the reference across all training data sizes. Finally, the training time for all the models is compared in Supplementary Table S1 showing that the PINN models converge faster than their corresponding data-driven models. Figure 6 visually compares the predictions by reference data-driven, PINN #1, and PINN #2 models developed with 70% and 5% training data shown for one complete shear stress and slip rate cycles. The superior performance of the PINN models for prediction of shear rate is visible in both panels and more pronounced for the smaller training set. In sum, our findings suggest that adding the physics constraints enhances the model performance and results in models with reasonable performance even when the training set is very small. Comparing PINN #1 and PINN #2 model performances for shear stress prediction, both PINN models show similar test scores except for the 50% and 5% cases where PINN # 2 outperforms PINN #1. The performance difference becomes more significant in the case of slip rate prediction with PINN #2 providing better predictions

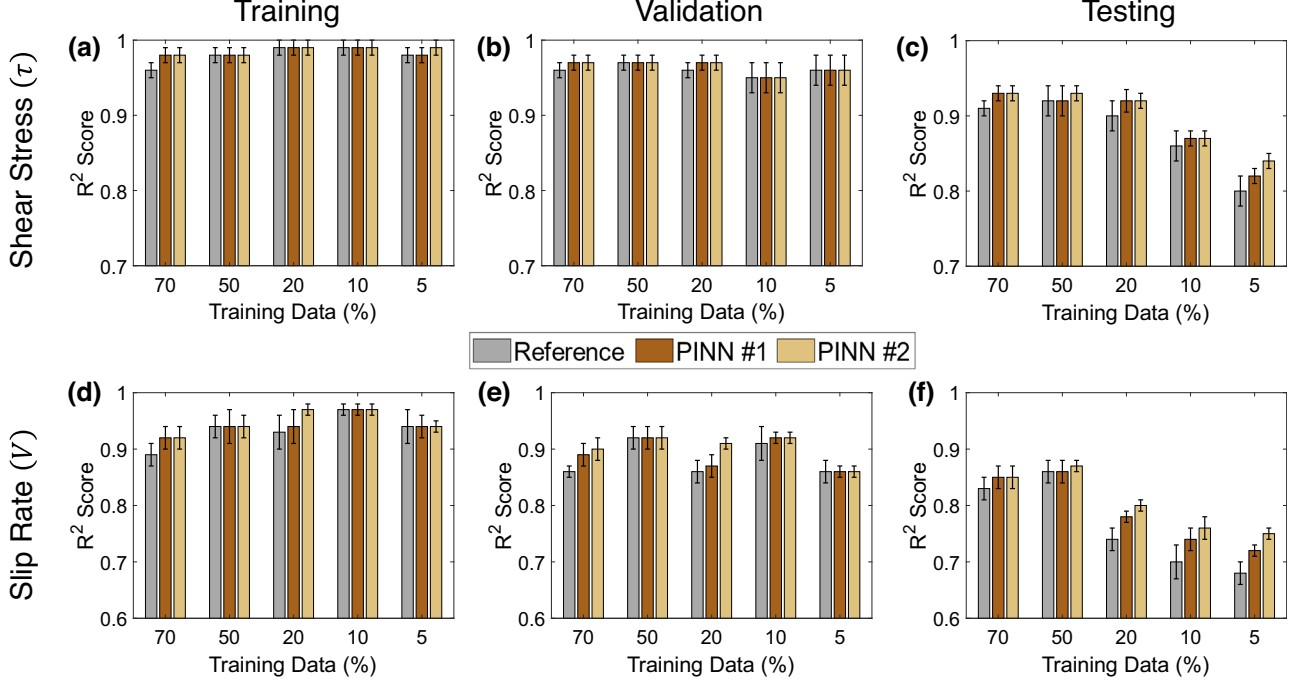

**Fig. 5 | Performance of the Reference data-driven, PINN #1, and PINN #2 models for experiment p5270. a–c** Shear stress ($\tau$) prediction $R^2$ scores in training, validation, and testing as a function of varying training set sizes. **d–f** Slip rate ($V$) prediction $R^2$ scores in training, validation, and testing datasets as a function of varying training set sizes are plotted. For both shear stress and slip rate, the PINN models outperform the reference data-driven models in testing and the improvement increases inversely with training data size. The minimum and maximum of the error bar represent the one standard error from the mean.

when ≤ 50% of the data are used for training. This highlights the importance of the constraint relating fault stiffness and ultrasonic wave transmission for improving prediction accuracy.

Finally, we examine how well the two PINN models have learned the experimental parameters and material properties in physics constraints (Eqs. (1)–(3)), which we treat as learnable constants. These weights are extracted from the learning layers of the fully trained models (early stopping enabled) and converted back to the original scale using the scaling used during data normalization. The learned values by PINN #1 & PINN #2 are then compared against the known values (when applicable) in Table 2. The PINN #1 model provides estimated normal stress ($\sigma$), system stiffness ($k$), and shear loading velocity ($v_l$) values with the percentage errors ranging from 2 to 14% compared to the true experimental values across all the varying training dataset sizes. Similarly, the PINN #2 model estimates the normal stress ($\sigma$), shear loading velocity ($v_l$), loading stiffness ($K$), and density ($\rho$) constants with smaller errors that range from 1 to 8%. In both cases, the error generally increases as the training set size decreases. The percent error for $A_{Intact}$ is not reported because its true value is not available. As expected from the performance comparison analysis above, constants estimated by PINN #2 models are more accurate than those estimated by PINN #1 models.

Transfer learning models are developed by fine-tuning data-driven and PINN models pre-trained on the p5270 dataset (70%–10%–20% split) to make predictions for the p5271 experiment. In addition, standalone data-driven models for p5271 experiment are trained, validated, and tested to serve as baseline. The training set size for the standalone and TL p5271 models is varied from 70% of the total data down to 10% while maintaining the same validation and testing sets of size 10% and 20%, respectively. Figure 7 compares the performance of the standalone and different TL models for all the considered data splits. Like for p5270, we see a generally decreasing $R^2$ score trend for standalone p5271 models when training data are reduced from 70% to 10%. We observe that all the TL models outperform standalone data-driven models. In most cases, the TL Data-driven and TL PINN #2 models show similar performances except for the 10% case, where the TL PINN #2 significantly outperforms the TL Data-driven. On the contrary, the TL PINN #1 models consistently outperform all the other models irrespective of data split. Further model tuning with cosine decay schedule and fine-tuning (freezing one or more layers) show that the TL PINN #1 models consistently outperform the TL PINN #2 models in predicting shear stress and slip rate in all scenarios. One possible reason behind this observation could be that the PINN #1 models are constrained only by the simplified elastic coupling relation (Eq. (1)) unlike PINN #2 which also incorporates the ultrasonic transmission involving ultrasonic attributes (Eqs. (1)–(3)) that vary from experiment to experiment. This could be why PINN #1 model may generalize better than PINN #2. Unlike p5270, we do not observe any consistent trends in the model variances. In general, all the TL and standalone models show a high variance in slip rate prediction compared to the shear stress prediction. In addition to $R^2$ score, the RMSEs for the standalone and TL models are compared in Supplementary Table S2. These corroborate the previous observations; all the TL models show smaller RMSE values compared to the standalone models with PINN #1 models having the smallest errors for each

**Table 1 | RMSE comparison between the reference, PINN #1, and PINN #2 models for experiment p5270**

| Model | Train-val-test | RMSE | | |
|---|---|---|---|---|
| # | (%) | Reference | PINN #1 | PINN #2 |
| 1 | 70-10-20 | 0.0923 | 0.0920 | 0.0900 |
| 2 | 60-10-20 | 0.0964 | 0.0943 | 0.0863 |
| 3 | 50-10-20 | 0.0986 | 0.0967 | 0.0880 |
| 4 | 40-10-20 | 0.0948 | 0.0938 | 0.0909 |
| 5 | 30-10-20 | 0.1080 | 0.1031 | 0.1030 |
| 6 | 20-10-20 | 0.1450 | 0.1269 | 0.1220 |
| 7 | 10-10-20 | 0.1480 | 0.1336 | 0.1227 |
| 8 | 5-10-20 | 0.1835 | 0.1654 | 0.1487 |

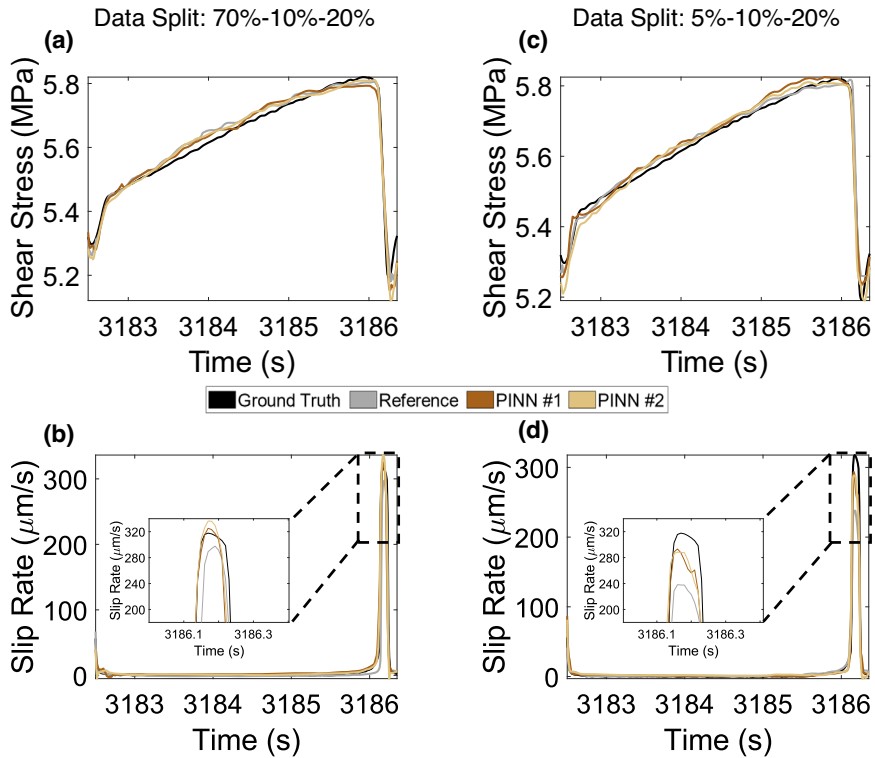

**Fig. 6 | A comparison of one shear stress (τ) & slip rate (V) cycle predicted by Reference (data-driven), PINN #1, and PINN #2 models. a, b** 70% of the data are used for training. **c, d** 5% of the data are used for training. The insets show that the PINN models perform better in predicting the slip rate compared to the data-driven models. The performance improvement is noticeable especially when the training data is scarce (5%).

**Table 2 | Experimental constants and material properties pertaining to p5270 experiment learned by: PINN #1 models & PINN #2 models**

| Train | PINN #1 | | | PINN #2 | | | | |
|---|---|---|---|---|---|---|---|---|
| (%) | $\sigma$ | $k$ | $v_l$ | $\sigma$ | $K$ | $v_l$ | $\rho$ | $A_{Intact}$ |
| 70 | 10.82 (+8%) | 0.0108 (−11%) | 8.50 (−5%) | 10.48 (+5%) | 0.0463 (−7%) | 8.7 (−3%) | 2619 (+1%) | 19893 |
| 60 | 10.93 (+9%) | 0.0107 (−13%) | 8.18 (−8%) | 10.63 (+6%) | 0.0463 (−7%) | 8.8 (−1%) | 2629 (+1%) | 19893 |
| 50 | 10.78 (+8%) | 0.0105 (−14%) | 8.60 (−3%) | 9.65 (−4%) | 0.0463 (−7%) | 8.8 (−1%) | 2625 (+1%) | 19893 |
| 40 | 10.44 (+4%) | 0.0110 (−10%) | 9.01 (+2%) | 9.89 (−1%) | 0.0463 (−7%) | 9.3 (+4%) | 2637 (+1%) | 19893 |
| 30 | 10.42 (+4%) | 0.0111 (−9%) | 9.17 (+3%) | 9.74 (−3%) | 0.0461 (−8%) | 8.7 (−3%) | 2639 (+2%) | 19893 |
| 20 | 10.46 (+5%) | 0.0110 (−10%) | 8.13 (−9%) | 9.85 (−2%) | 0.0461 (−8%) | 8.5 (−4%) | 2637 (+1%) | 19893 |
| 10 | 10.30 (+3%) | 0.0110 (−10%) | 8.19 (−8%) | 9.86 (−1%) | 0.0465 (−7%) | 8.8 (−1%) | 2683 (+3%) | 19893 |
| 5 | 10.36 (+4%) | 0.0110 (−10%) | 7.89 (−11%) | 9.94 (−1%) | 0.0461 (−8%) | 8.7 (−2%) | 2643 (+2%) | 19893 |
| Actual value | 10 ± 0.01 | 0.0122 | 8.9 | 10 ± 0.01 | 0.050 | 8.9 | 2600 | N/A |

The numbers in parentheses show the percentage deviation of the learned constants from the known corresponding values. In all models, the validation and testing dataset is set to 10% and 20% of the entire dataset. Parameters and their units: $\sigma$ (MPa), $k$ (MPa/μm), $v_l$ (μm/s), $K$ (MPa/μm), $\rho$(kg/m³) & $A_{Intact}$ (arb. units).

split. Comparing the time required for training in Supplementary Table S3, the TL models are shown to converge faster, indicating that the initial weights from the p5270 model provide a good starting point for training p5271 models. Finally, the experimental constants learned by TL PINN #1 and TL PINN #2 frameworks are compared against their known values in Supplementary Table S4. In sum, TL improves model prediction. Moreover, when the training data are scarce (represented by 10% of training data) the TL PINN models outperform standalone and TL data-driven models by a large margin.

**Relevance to field studies**

This study demonstrates that adding physics-based constraints to ML models is greatly beneficial for failure prediction, especially when datasets are scarce. On the other hand, we recognize that the model

developed here cannot be directly applied to field data, because shear stress and slip rate data at depth are not accessible in the field. Moreover, very few active seismic surveys performed continuously over extended periods of time are available[58]. Nonetheless, we believe this work represents a step toward failure prediction in the field for the following reasons. First, an approach similar to the one presented here still using lab data might be followed to better constrain the rate and state frictional models and associated parameters that are used in geodetic studies to infer fault slip distribution at depth[59–67]. Second, stress and slip rate data might be inferred in the field with ML models using earthquake recurrence as input data, and possibly pre-training on lab data.

We build a PINN framework to predict laboratory earthquakes from active-source ultrasonic monitoring data and demonstrate that

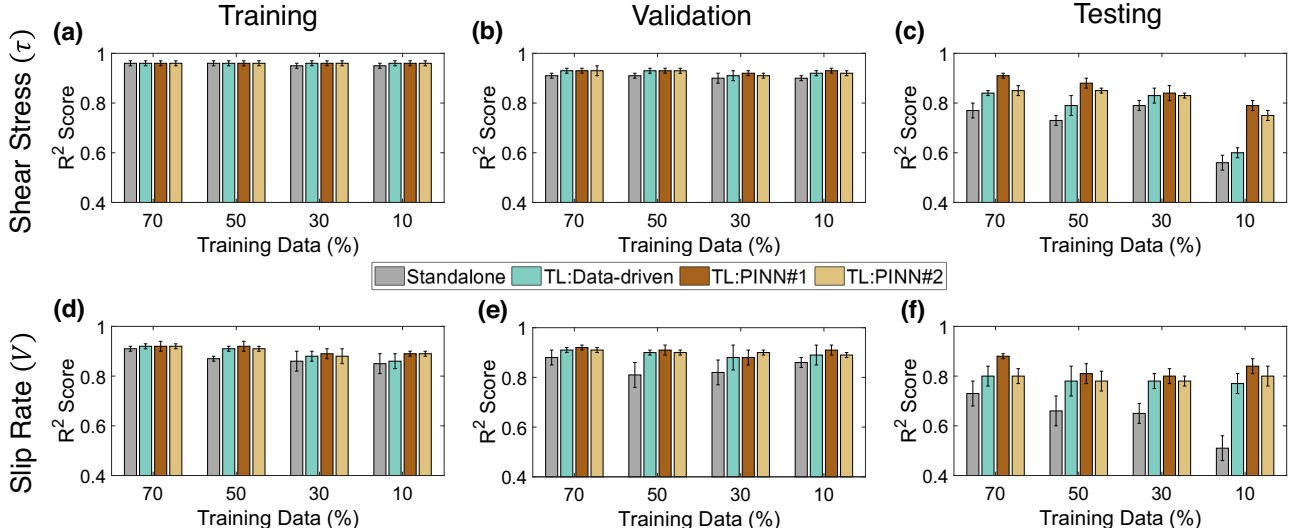

**Fig. 7 | Performance of the Standalone, TL: data-driven, TL PINN #1 and TL: PINN #2 models for experiment #p5271.** TL models are initialized using the p5270 dataset. **a–c** Shear stress ($\tau$) prediction $R^2$ scores in training, validation, and testing datasets as a function of varying training set sizes are plotted. **d–f** Slip rate ($V$) prediction $R^2$ scores in training, validation, and testing datasets as a function of varying training set sizes are plotted. TL improves model prediction and the TL PINN models significantly outperform standalone and TL data-driven models for small size training sets. The minimum and maximum of the error bar represent the one standard error from the mean.

it outperforms models based only on data-driven loss especially when training data are limited. This framework incorporates two physical constraints that describe the elastic coupling of faults with their surroundings as well as ultrasonic transmission across the frictional interface. We compare the PINN predictions of laboratory earthquakes (shear stress history) and slip rate evolution to those from the purely data-driven models for varying amounts of training data. A key result is that incorporating the physics constraints improves the model performance; the improvement is most significant when training data are scarce. The modeling results for one dataset (p5270) suggest that the PINN #2 framework (constrained by both elastic coupling and ultrasonic transmission relations) outperforms the PINN #1 framework (incorporating only elastic coupling of faults with their surroundings). Furthermore, a TL study carried out using a distinct second dataset (p5271) shows that TL models outperform standalone models and that with transfer-learned PINN, it is possible to develop reasonably well-performing prediction models using a small amount of training dataset (using 10% in this study). In sum, our findings suggest that incorporating simplified laws of physics results in accurate and transferable predictions even when the training data size is small. This finding has important implications for seismicity monitoring and prediction in $CO_2$ storage sites, geothermal and unconventional reservoirs using the time-lapsed active source monitoring with limited available field data.

## Methods

Our objective is to predict shear stress ($\tau_i$) history (model output) given the time evolution of the extracted ultrasonic features, $v_i$ and $A_i$ (input features). In addition to ($\tau_i$), we also predict slip rate ($V_i$) to formulate one of the physics constraints. Dual-output data-driven DL and PINN models are developed to simultaneously predict shear stress $\tau_i$ and slip rate $V_i$ histories, using the time-evolution of wave speed $v_i$ and amplitude $A_i$ features as input. For both sets of models, we implement MLP which is a deep fully connected neural network structure to perform the regression task. Note that time to failure (TTF) is not directly predicted here as it is not independent of shear stress ($\tau_i$). If desired, the TTF for each stick-slip cycle can be readily estimated from the predicted $\tau_i$ history.

### Training-validation-testing splits

To build the models, the dataset from each experiment is divided into distinct training, validation, and testing sets. The models are trained on the training dataset; hyperparameter tuning is carried out on the validation dataset while the unseen testing dataset is used to evaluate the reported performance of the models. Because we use time series of continuous data, the data are not randomly sampled or shuffled during training, validation, or testing preserving the sequence of stick-slip cycles. To investigate the prediction performance of the models with limited training data, the amount of training data used is varied from 70% (equivalent to about 273 seismic cycles) down to 5% (equivalent to about 15 seismic cycles) of the entire dataset, whereas the same validation and testing datasets are used across the models. The validation and testing sets constitute the final portion of the data and amount to 10% and 20% of the dataset, respectively. Note that we choose the training, validation, and testing sets in a sequence i.e., the training dataset immediately precedes the validation set, which is followed by the testing set as shown in Fig. 8.

### Normalization

Prior to building the models, all the data are scaled using min-max normalization. This is achieved by first normalizing the training data followed by normalizing the validation and test datasets using the same training data min-max values.

### Data-driven models

Data-driven MLP models are developed as reference models for all the training-validation-test splits. An MLP model consists of an input layer, one or more hidden layers, and an output layer. The data are propagated forward from the input layer to the output layer and the neurons are trained with the backpropagation learning algorithm[56]. Through grid search, we explored a series of MLP models with a different number of layers, nodes, batch sizes, and learning rates to find the best hyperparameters based on the performance on the validation dataset. Our best-performing data-driven MLP model has five hidden layers with 128, 64, 32, 16, and 8 nodes, respectively. A batch size of 32 and a learning rate of 0.001 is used following hyperparameter tuning.

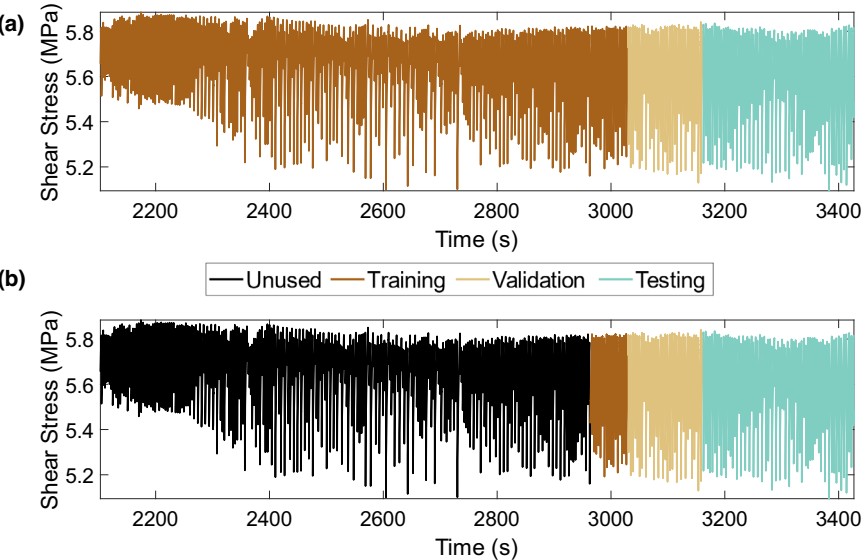

**Fig. 8 | Data from experiment p5270 showing shear stress history and two different training-validation-test splits. a** 70% of the data (equivalent to 273 seismic cycles) are used for training. **b** 5% of the data (equivalent to 15 seismic cycles) are used for training. Note that in both cases, the validation and test sets do not vary and consist of 10% and 20% of the data equivalent to 36 and 78 cycles, respectively. Therefore, the total amount of data used for developing the model is varied from 100% (= 70 + 10 + 20) down to 35% (= 5 + 10 + 20) corresponding to the respective splits shown in (**a**, **b**).

Following ref. [42], the input features are provided with 3 s of data history before the current time to predict shear stress and slip rate at the current time. The 3 s correspond roughly to the average duration of a seismic cycle in our datasets. We use mean squared error (MSE) loss, rectifier linear unit (ReLU) activation for each hidden layer, and linear activation for the output layer. The number of epochs is set to 100 with early stopping (patience = 20) enabled to prevent overtraining. Finally, we use the Adam optimizer[57] and the $R^2$ score metric to evaluate model performance.

## PINN modeling framework

The proposed PINN framework builds upon the data-driven MLP model discussed above. We modify the loss function to include physics-based constraints as illustrated schematically in Fig. 3. We consider two constraints. The first describes elastic coupling between a fault and its surroundings. We model the lab setup as a single-degree-of-freedom spring slider system neglecting inertia[52]:

$$\frac{d\mu}{dt} = \frac{d\left(\frac{\tau_i}{\sigma}\right)}{dt} = k\left(V_l - V_i\right) \tag{1}$$

where $\mu (= \tau_i/\sigma)$ is the coefficient of friction and $k$ is the overall system stiffness. Shear stress $\tau_i$ and slip rate $V_i$ are the model outputs while the applied normal stress $\sigma$ and shearing rate $V_l$ are known experimental constants. The system stiffness $k$ combines the fault stiffness $k_f$ and the stiffness $K$ of the rest of the deformation machine and host rock loading the fault (a measurable experimental parameter), which act in parallel:

$$\frac{1}{k} = \frac{1}{K} + \frac{1}{k_f}. \tag{2}$$

The fault stiffness $k_f$ is related to the ultrasonic transmission coefficient $T$ across the fault interface through the displacement discontinuity model[54]:

$$T = \frac{A_i}{A_{Intact}} = \frac{2k_f}{\sqrt{\left(2\pi f_0 \rho v_i\right)^2 + 4k_f^2}} \tag{3}$$

where $A_i$ is the transmitted wave amplitude history, $A_{intact}$ is the transmitted wave amplitude through the intact granite blocks i.e., in the absence of the faults, $f_0$ is the center frequency of the received wave (400 kHz), $\rho$ is the mass density of the material surrounding the fault (granite), and $v_i$ is the wave speed history. Among these, $A_i$ and $v_i$ are the input features of the model while $A_{intact}$ and $\rho$ are experimental constants. The center frequency history $f_0$ could have been extracted from the ultrasonic signals and used as an input, but here, we opt to treat it as a given constant $f_0$. Note that Eq. (2) couples Eqs. (1) and (3) through $k_f$.

In this study, we consider two different PINN models: PINN #1 and PINN #2. The first one (PINN #1) is only constrained by the elastic coupling relation (Eq. (1)) that includes a relation between the two output variables $\tau_i$ and $V_i$ (but no input features) rewritten as:

$$f_1 : \frac{d\left(\frac{\tau_i}{\sigma}\right)}{dt} - k\left(V_l - V_i\right) = 0 \tag{4}$$

The second PINN model (PINN #2) is constrained by Eqs. (1) to (3) and therefore, also includes the constraint involving input features wave speed $v_i$ and wave amplitude $A_i$. To build the PINN #2 model, all three equations are combined to produce the following constraint added to the loss function:

$$f_2 : \frac{d\left(\frac{\tau_i}{\sigma}\right)}{dt} - \frac{\pi f_0 \rho v_i T K}{\pi f_0 \rho v_i T + K\sqrt{1 - T^2}}\left(V_l - V_i\right) = 0 \tag{5}$$

Given that shear stress $\tau_i$ and slip rate $V_i$ are the model outputs, we denote that the predicted shear stress and slip rate as $\hat{\tau}_i$ and $\hat{V}_i$, respectively. We can view these predictions as functions of a time-dependent input feature vector $(u_i = A_i, v_i)$ and $\theta$, which are a collection of weight matrices and bias vectors used by the model. By substituting the neural network approximations into the governing equations (Eq. (4)) and (Eq. (5)), we obtain the constraint functions $\hat{f}_1$ (Eq. (6)) and $\hat{f}_2$ (Eq. (7)) below:

$$\hat{f}_1 = \frac{d\left(\frac{\hat{\tau}_i(u_i;\theta)}{\sigma}\right)}{dt} - k\left(V_l - \hat{V}_i(u_i;\theta)\right) \tag{6}$$

$$\hat{f}_2 = \frac{d\left(\frac{\hat{\tau}_i(u_i;\theta)}{\sigma}\right)}{dt} - \frac{\pi f_0 \rho v_i T K}{\pi f_0 \rho v_i T + K\sqrt{1-T^2}}\left(V_l - \hat{V}_i(u_i;\theta)\right) \quad (7)$$

Finally, the composite cost functions for PINN #1 and PINN #2 framework are written in Eqs. (8) and (9), respectively.

$$L_1(\theta) = \frac{1}{N}\sum_{i=1}^{N}\left(\tau_i(u_i) - \hat{\tau}_i(u_i;\theta)\right)^2 + \frac{1}{N}\sum_{i=1}^{N}\left(V_i(u_i) - \hat{V}_i(u_i;\theta)\right)^2 + \frac{1}{N}\sum_{i=1}^{N}\left(\hat{f}_1(u_i;\theta)\right)^2 \quad (8)$$

$$L_2(\theta) = \frac{1}{N}\sum_{i=1}^{N}\left(\tau_i(u_i) - \hat{\tau}_i(u_i;\theta)\right)^2 + \frac{1}{N}\sum_{i=1}^{N}\left(V_i(u_i) - \hat{V}_i(u_i;\theta)\right)^2 + \frac{1}{N}\sum_{i=1}^{N}\left(\hat{f}_2(u_i;\theta)\right)^2 \quad (9)$$

The cost function combines the data-driven and physics costs using a regularizer value of 1. For both models, the first two terms represent the MSE in predicting the shear stress ($\tau_i$) and slip rate ($V_i$) histories. The $\hat{f}$ terms represent the penalty added to the cost function for violating the physics constraint defined using Eqs. (1)–(3). The other experimental parameters in the physics equations (i.e., $\sigma, K, V_l, \rho, A_{Intact}$) are neither model inputs nor the target outputs. Although these parameters are either known ($\sigma, V_l, \rho, k$) or measurable ($A_{Intact}$ could be measured by testing an intact granite block of the same thickness as the cumulative thickness of the blocks used in the friction experiment), we treat them as trainable neural network weights in the PINN framework. This approach is used to avoid errors due to unit mismatch between features, outputs, and these constants in the constraints. These weights are extracted from the layers of the fully trained models and converted back to the original scale to undo the effect of data normalization (see implementation details in https://github.com/prabhavborate92/PINNPaper.git ). A comparison of the scaled learned weights with the known parameter values gives us the opportunity to examine the PINN model by determining how well the models are able to learn the values of parameters measured experimentally.

### Transfer learning

As a way to assess the models' generalizability, we use a transfer learning approach i.e., apply each of the three models (purely data-driven, PINN 31, and PINN 62) trained on the p5270 (reference model) to a new experiment: p5271. Figure 4 illustrates the proposed transfer learning approach. Transfer learning is carried out by using the same neural network architecture but fine-tuning (further training) of all the pre-trained reference model weights (trained and validated on the p5270 dataset) as well as the learning rate and batch size using only a small training set from experiment p5271. In other words, instead of random initialization, the weights of the new model are initialized using what has been learned when training the reference model. These initial weights and hyperparameters provide a good starting point for building a model on the new dataset and lead to faster convergence and smaller training set sizes. For all the transfer-learned (TL) models, hyperparameter tuning resulted in a learning rate of 1e-3 and batch size of 32. Similar to the data-driven models the TL models are trained with the number of epochs set to 100 and using Adam as an optimization algorithm.

We compare the performance of the models after transfer learning with that of the standalone data-driven model trained solely on the p5271 dataset with the training dataset size varied between 70 and 10% of the entire dataset (similar to that explained for standalone models trained on p5270 experimental dataset). Finally, the performance of

the standalone and transfer-learned (data-driven, PINN 91, and PINN 32) models are compared in terms of $R^2$ scores (as the performance metric), RMSE and required training time for each model.

## Data availability

The experiment p5270 and p5271 data used for training, validation, and testing can be found at: https://github.com/prabhavborate92/PINNPaper.git .

## Code availability

The source codes and models developed in this paper can be accessed at https://github.com/prabhavborate92/PINN_Paper.git. When using the codes and models available in the GitHub repository, please cite Borate et al.[68].

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

## Acknowledgements

This study is funded by the US Department of Energy (DE-SC0017585) and NSF-MCA (#2121005) grants to P.S. C.M. acknowledges support from the European Research Council Advance Grant 835012 (TECTONIC) and US Department of Energy grants (DE-SC0020512) and (DE-EE0008763). J.R. acknowledges support from the US Department of Energy grant (DE-SC0022842). The authors are thankful to Srisharan Shreedharan for providing his experiment datasets and Prabhakaran Manogharan for his help in extracting the ultrasonic features.

## Author contributions

P.B.: methodology and writing—original draft. J.R.: writing—review & editing, and supervision. C.M.: writing—review and editing. A.M.: methodology and writing—review. D.K.: methodology and writing—review. P.S.: conceptualization, methodology, writing—review & editing, supervision, project administration, and funding acquisition.

## Competing interests

The authors declare no competing interests.
