## [Peer Review File · Nature Communications]

Using a physics-informed neural network and fault zone acoustic monitoring to predict lab earthquakesREVIEWER COMMENTS

Reviewer #1 (Remarks to the Author):

Reviews to

Using a physics-informed neural network and
fault zone acoustic monitoring to predict lab earthquakes

by Prabhav Borate, Jacques Riviere, Chris Marone, Ankur Mali, Daniel Kifer, and Parisa Shokouhi

Summary: The authors have developed a workflow to design and train a neural network to predict shear stress and slip speed on a laboratory fault using physics-based regularization terms in the loss function. They found that the addition of physics-based regularization helped find more generalizable models (perform better on test data set) that still perform well when few training data are available. The manuscript is well written, the results are clearly presented and the benefits of using physics-based regularization are clearly demonstrated. The data and codes used in the study are publicly available online. My comments are only about the message delivered by the paper, I don't think that any additional technical work is necessary.

Major comments:

- The authors introduce their work in the context of the great challenge of predicting failure in the field (either for natural, tectonic earthquakes or induced earthquakes) and claim that this study will help design future models for field applications. While I'm fully convinced that such physics-based regularizers should be included as much as possible in geophysical studies, I think that the authors should elaborate more on how their study relates to a possible application in the field. The model uses two variables, the shear stress and slip speed on the fault, that are completely unavailable in the field with today's technology. Getting the shear stress and slip speed in space and time on natural faults is certainly a much harder problem than using these variables to predict the next failure and a neural network might then not even be necessary to anticipate failure. In its current shape, I find the study to be more a proof that failure is predictable in a relatively simple rock sample than an actual step towards predicting failure on natural faults.

- The answer to this question might be in the "Data-Driven Models" section but I wasn't sure: For a given size of the training set, do you use the same MLP architecture for all three models? It seems to be an important point because if you repeat the hyperparameter tuning and find different optimal architectures for the data-driven vs PINN 1 vs PINN 2 model, then you are not only comparing different loss functions but also different architectures, which complicate the interpretation of the comparison.

Minor comments:

- Lines 32-35: The authors use the term "acoustic emissions" to talk equally about discrete events and the continuous recordings, which I find confusing. It gives the impression that 100% of the signal in the continuous recordings is due to failure events whereas part of it is "ambient noise". Although "ambient noise" in the lab might very well mostly be caused by the reverberations of the waves emitted by the AE events, I think it's important to note that waves that are not radiated from the monitored fault can carry relevant information about the elastic properties of the bulk around the fault. In that sense, I think the two sentences lines 32-35 are misleading because they say that hidden, active sources carry information whereas "ambient noise" also carries relevant information. In other terms, the relevant information might not be that much in the source signature than in the path signature. It's probably me not interpreting well the sentences but I suggest making the phrasing clearer as I might not be the only one reading it this way.

Reviewer #2 (Remarks to the Author):

Dear Authors and Editor,

I am writing to provide my reviews for the paper NCOMMS-22-50565-T titled "Using a physics-informed neural network and fault zone acoustic monitoring to predict lab earthquakes" by Prabhav Borate et al.

The paper describes how physics-informed neural networks can be used to improve the regression between experimental observations (ultrasonic wavespeed and transmission amplitude) and the shear stress and slip rate of a loaded block analog to natural fault producing lab quakes. I found the experimental setup and data processing stages to be clearly described and of high quality. Importantly, the authors address the question of how physics-based constraints can be used to improve machine-learning based regression. The authors show that integrating physics-based knowledge into the neural network can enhance its performance in comparison to classical multilayer perception models, particularly when dealing with limited training data. This finding is particularly relevant for potential applications to field data, where multiple seismic cycles are not always observed. Furthermore, they highlight the tradeoff between imposing too much a priori knowledge versus providing enough flexibility to generalize the results, with a transfer learning experiment.

Overall, I believe the paper is of high quality and has the potential to make a significant contribution to the field. With some minor corrections, I recommend that it should be considered for publication. Please find below my general remark, which I hope will be useful to the authors in revising their work.

Regarding my general remark, I would like to draw your attention to the observation made in the transfer learning experiment, where the less constrained PINN outperforms the most constrained one. While the authors provide a sentence to discuss this result, I think it may deserve more insights. As stated by the authors, the likely explanation for this is the too much constrained loss function in PINN#2, which incorporates ultrasonic attributes and may not be transferable straightforwardly between experiments. However, I do not fully understand why this point is not tackled by the fine-tuning performed in the transfer learning approach. In other words, would this point be better addressed by retraining PINN#2 sufficiently or performing a larger fine-tuning, or is the network stuck in a local minimum?

In terms of minor comments, I suggest capitalizing "physics-informed neural network" to emphasize the definition of the acronym PINN, both at line 14 and when it is redefined at line 49. Also, "SHM" is defined as an acronym but only used once; therefore, you may consider removing it at line 17.

Regarding Figure 1a, the shear stress and slip rate are encoded with the same color, and the line styles do not allow distinguishing them. Would it be possible to differentiate them with two colors? Additionally, in Figure 1b, you may consider using the label "Normalized shear stress and slip rate" instead of "Normalized", and the "Time" label may also refer to "Ultrasonic signal local time".

In the Ultrasonic Feature Extraction section, could you please clarify how you select the reference waveform, and whether your results are reference-dependent?

Finally, PZT is defined but used only once at line 102.

Thank you for the opportunity to review this manuscript. Please do not hesitate to contact me if you have any questions or require any further information.

Sincerely,

Reviewer #3 (Remarks to the Author):

Comments for Borate et al. "Using a physics-informed neural network and fault zone acoustic monitoring to predict lab earthquakes "

This study shows the superiority of physics-informed neural networks (PINN) compared to data-driven neural networks for predicting shear stress history and slip rates in laboratory shear-slip experiments. The structure of the paper is straightforward and clearly demonstrates the superiority of PINN, and I did not find any particular problem. The manuscript is clearly written, and I personally think it is almost ready for publication. One concern is that the comparison target is a simple multilayer perceptron neural network that is not expected to perform well, so the paper is not very interesting in terms of performance. However, this study shows that PINN, which includes simple physical rules, can be effective to a certain extent, and I consider this study worth publishing.

P.3 L 104: Are the piezoelectric transducers used as a transmitter and a receiver the same type? If possible, it would be desirable to specify the manufacturer and model number.

P. 4 L 123: Does this 10-point backward-looking average affect the results?

P. 10 L 301 I did not understand how the parameters of σ , K , ... were obtained in the PINN approach. Could you add the description?

Fig. 1 It is difficult to distinguish shear stress and slip rate in this figure. I want to suggest using different colors.

The authors used a simple perceptron model as a data-driven model. In my opinion, for such time series data, the convolutional neural network approach can obtain better performance, possibly comparable to PINN results. Are the authors performed such validation?

Response to reviewers

The authors thank the reviewers for their insightful comments, which have improved and clarified the manuscript. The revisions made to the manuscript in response to the reviewers' comments are all noted in **red** for your reference. Below are our answers to all the comments. The reviewers' comments are written in **black**, and our responses are in **blue**.

Reviewer #1 (Remarks to the Author):

Summary: The authors have developed a workflow to design and train a neural network to predict shear stress and slip speed on a laboratory fault using physics-based regularization terms in the loss function. They found that the addition of physics-based regularization helped find more generalizable models (perform better on test data set) that still perform well when few training data are available. The manuscript is well written, the results are clearly presented and the benefits of using physics-based regularization are clearly demonstrated. The data and codes used in the study are publicly available online. My comments are only about the message delivered by the paper, I don't think that any additional technical work is necessary.

Major comments:

1. The authors introduce their work in the context of the great challenge of predicting failure in the field (either for natural, tectonic earthquakes or induced earthquakes) and claim that this study will help design future models for field applications. While I'm fully convinced that such physics-based regularizers should be included as much as possible in geophysical studies, I think that the authors should elaborate more on how their study relates to a possible application in the field. The model uses two variables, the shear stress and slip speed on the fault, that are completely unavailable in the field with today's technology. Getting the shear stress and slip speed in space and time on natural faults is certainly a much harder problem than using these variables to predict the next failure and a neural network might then not even be necessary to anticipate failure. In its current shape, I find the study to be more a proof that failure is predictable in a relatively simple rock sample than an actual step towards predicting failure on natural faults.

We completely agree with the reviewer that such an approach is not readily applicable at this time, but we would like to promote a broader view of physics-informed approaches. We have added a paragraph in the discussion section (lines 230-240) to clarify this aspect and provide arguments as to why this study nonetheless represents a step toward prediction in the future.

"This study demonstrates that adding physics-based constraints to ML models is greatly beneficial for failure prediction, especially when datasets are scarce. On the other hand, we recognize that the model developed here cannot be directly applied to field data, because shear stress and slip rate data at depth are not accessible in the field. Moreover, very few active seismic surveys performed continuously over extended periods of time are available (Niu et al., 2008). Nonetheless, we believe this work represents a step toward failure prediction in the field for the following reasons. First, an approach similar

to the one presented here still using lab data might be followed to better constrain the rate and state frictional models and associated parameters that are used in geodetic studies to infer fault slip distribution at depth (Avouac, 2015; Barbot et al., 2012; Bürgmann et al., 2002; Fielding et al., 2013; Gualandi et al., 2017; Hearn & Bürgmann, 2005; Kano et al., 2018; Michel et al., 2019; Wallace et al., 2016). Second, stress and slip rate data might be inferred in the field with ML models using earthquake recurrence as input data, and possibly pre-training on lab data.”

2. The answer to this question might be in the “Data-Driven Models” section but I wasn’t sure: For a given size of the training set, do you use the same MLP architecture for all three models? It seems to be an important point because if you repeat the hyperparameter tuning and find different optimal architectures for the data-driven vs PINN 1 vs PINN 2 model, then you are not only comparing different loss functions but also different architectures, which complicate the interpretation of the comparison.

Yes, for a given size of the training data split, all the models (data-driven, PINN 1, and PINN 2) use the same MLP architecture. The PINN 1 and PINN 2 frameworks are developed based on the data-driven models, therefore all three models share the same architecture, including the number of hidden layers and units, batch size, optimizer, and learning rate. To clarify these points, the following sentence is added to the revised manuscript in the Results and Discussion section (lines 141-143).

“The data-driven, PINN 1, and PINN 2 models share the same MLP framework (hidden layers, units, batch size, optimizer, and learning rate) across different data splits to allow one on one comparison.”

Minor comments:

1. Lines 32-35: The authors use the term “acoustic emissions” to talk equally about discrete events and the continuous recordings, which I find confusing. It gives the impression that 100% of the signal in the continuous recordings is due to failure events whereas part of it is “ambient noise”. Although “ambient noise” in the lab might very well mostly be caused by the reverberations of the waves emitted by the AE events, I think it’s important to note that waves that are not radiated from the monitored fault can carry relevant information about the elastic properties of the bulk around the fault. In that sense, I think the two sentences lines 32-35 are misleading because they say that hidden, active sources carry information whereas “ambient noise” also carries relevant information. In other terms, the relevant information might not be that much in the source signature than in the path signature. It’s probably me not interpreting well the sentences but I suggest making the phrasing clearer as I might not be the only one reading it this way.

We agree with the reviewer that this paragraph needs to be corrected and clarified. The passive data collected during such experiments are recorded continuously and contain impulsive events, as well as a featureless signal that looks like noise. Our past work suggests that this featureless signal can be either due to a lack of events (locked fault), or

many small AE events that overlap and cannot be isolated. The latter case was encountered when attempting to catalog events (Bolton et al., 2020), where we find that there is a deficit of small events at such low amplitudes (and this magnitude of completeness is found to increase with the imposed shearing velocity). From our current work and work in progress (Marty et al., in preparation), we know that the vast majority of events occur on the fault plane, but we agree that the energy captured by the sensors also comes from subsequent scattering, carrying information about the stress state within the host rock. The paragraph now reads as follows (lines 28-40 of the revised manuscript):

“Numerous laboratory studies have shown that the onset of failure is associated with bursts of acoustic emission (AE) events taking place during crack initiation and growth, and the number and amplitude of AE events generally increase as the sample approaches failure (Bolton et al., 2019; Bu et al., 2022; Chow et al., 1995; Dunegan & Harris, 1969; Jansen et al., 1993; Lockner, 1993; Rivière et al., 2018; Rouet-Leduc et al., 2017; Savage & Hasegawa, 1964; Scholz, 1968). Recent friction studies on laboratory faults have shown that machine learning (ML) algorithms can actually predict the timing and magnitude of lab quakes using AE data (Bolton et al., 2019; Hulbert et al., 2019; Jaspersen et al., 2021; Laurenti et al., 2022; Lubbers et al., 2018; Pu et al., 2021; Rouet-Leduc et al., 2017, 2018; Wang et al., 2021, 2022). It is remarkable that solely using acoustic emission data radiating from the faults as an input, the fault strength can be accurately predicted throughout the laboratory seismic cycle (Hulbert et al., 2019; Rouet-Leduc et al., 2018). Past work (Blanke et al., 2021; Goebel et al., 2012, 2013) has shown that the vast majority of events radiate from the fault plane, therefore carrying information about the fault state. And as the elastic waves radiate/scatter through the host granite blocks, they also provide information about the stress state of the host rock. It is also remarkable that predictions work in the early stage of the seismic cycle when the acoustic signal often looks like noise, either because it lacks a clear P-wave, such as expected for friction/fracture events, or because it represents something like tectonic tremor involving the sum of many small or low frequency events that overlap in time and cannot be distinguished as separate events (Bolton et al., 2020).”

Reviewer #2 (Remarks to the Author):

Dear Authors and Editor,

I am writing to provide my reviews for the paper NCOMMS-22-50565-T titled "Using a physics-informed neural network and fault zone acoustic monitoring to predict lab earthquakes" by Prabhav Borate et al.

The paper describes how physics-informed neural networks can be used to improve the regression between experimental observations (ultrasonic wavespeed and transmission amplitude) and the shear stress and slip rate of a loaded block analog to natural fault producing lab quakes. I found the experimental setup and data processing stages to be clearly described and of high quality. Importantly, the authors address the question of how physics-based constraints can be used to improve machine-learning based regression. The authors show that integrating physics-based knowledge into the neural network can enhance its performance in comparison to classical multilayer perception models, particularly when dealing with limited training data. This finding is particularly relevant for potential applications to field data, where multiple seismic cycles are not always observed. Furthermore, they highlight the tradeoff between imposing too much a priori knowledge versus providing enough flexibility to generalize the results, with a transfer learning experiment.

Overall, I believe the paper is of high quality and has the potential to make a significant contribution to the field. With some minor corrections, I recommend that it should be considered for publication. Please find below my general remark, which I hope will be useful to the authors in revising their work.

1. Regarding my general remark, I would like to draw your attention to the observation made in the transfer learning experiment, where the less constrained PINN outperforms the most constrained one. While the authors provide a sentence to discuss this result, I think it may deserve more insights. As stated by the authors, the likely explanation for this is the too much constrained loss function in PINN#2, which incorporates ultrasonic attributes and may not be transferable straightforwardly between experiments. However, I do not fully understand why this point is not tackled by the fine-tuning performed in the transfer learning approach. In other words, would this point be better addressed by retraining PINN#2 sufficiently or performing a larger fine-tuning, or is the network stuck in a local minimum?

Thank you for this comment. Besides hyperparameter tuning carried out in our study, we present the results for cosine decay, and layer freezing in our transfer learning (TL) study to confirm whether such a detailed fine-tuning can address this point. Selecting an appropriate learning rate during model training is crucial for optimizing the weights. Optimization diverges if the learning rate is set too high and if it is set too low, convergence is slow. For all the transfer learned (TL) models, hyperparameter tuning using grid search resulted in finding an optimal learning rate of $1e-3$ and batch size of 32, as reported in the originally submitted manuscript in lines 317 to 319.

Cosine decay is one of the most widely used approaches for learning rate decay, which help the model move away from the saddle point and helps in converging and reaching better local minima. Cosine decay including other variants such as polynomial decay is commonly used to stably train transformers and other SoTA models. As shown in Figure 1 below, we started the model with a learning rate of 10^{-2} and gradually decreased it to 10^{-4} as training progressed. Table 1 below illustrates the performance of the PINN #1 and PINN #2 models using this decay for varying training data sizes. Consistent with the results presented in the paper, PINN #1 models outperform PINN #2 models with consistently higher R^2 scores and align with the results shown in our manuscript.

Figure 1: Model training with a varying learning rate from 10^{-2} to 10^{-4} following the cosine decay schedule.

Table 1: PINN #1 and PINN #2 model performance with varying training dataset sizes for predicting shear stress and slip rate with cosine decay schedule.

Training Data (%)	Shear Stress R^2		Slip Rate R^2	
	TL: PINN #1	TL: PINN #2	TL: PINN #1	TL: PINN #2
70	0.84	0.82	0.8	0.78
50	0.86	0.79	0.82	0.67
30	0.77	0.77	0.73	0.67
10	0.72	0.51	0.70	0.57

The PINN models developed have 5 hidden layers. For the fine-tuning study, the models trained on the p5270 dataset have K frozen (not trainable) layers and (N-K) trainable layers, where N is the total number of layers and K is the total number of frozen layers. We use (N-K) layers to optimize our model on the p5271 dataset, to better understand the feature importance that is learned on the prior dataset. Table 2-5 compares the performance of the PINN #1 and PINN #2 models for this investigation on the testing dataset with 70%-10% training dataset respectively. Again, the PINN #1 models consistently outperform the PINN #2 models in predicting shear stress and slip rate in all scenarios of fine-tuning. The models perform best when all of the layers are unfrozen (allowed to train); and all model weights are initialized using the pre-trained network, such that fine-tuning them with a low learning rate leads to better performance, on the other hand, performance starts degrading as (N-K)(number of the trainable or unfrozen layer) < K (frozen layer).

Table 2: PINN #1 and PINN #2 model performance with 70% training dataset for predicting shear stress and slip rate with fine-tuning.

Frozen Layer (K)	Shear Stress R ²		Slip Rate R ²	
	TL: PINN #1	TL: PINN #2	TL: PINN #1	TL: PINN #2
None or K=0	0.91	0.85	0.88	0.8
1 or K=1	0.77	0.68	0.50	0.45
1-2 or K=2	0.77	0.58	0.48	0.47
1-3 or K=3	0.71	0.61	0.48	0.41
1-4 or K=4	0.59	0.46	0.41	0.30

Table 3: PINN #1 and PINN #2 model performance with 50% training dataset for predicting shear stress and slip rate with fine-tuning.

Frozen Layer	Shear Stress R ²		Slip Rate R ²	
	TL: PINN #1	TL: PINN #2	TL: PINN #1	TL: PINN #2
None or K=0	0.88	0.85	0.81	0.78
1 or K=1	0.81	0.62	0.55	0.35
1-2 or K=2	0.79	0.65	0.58	0.48
1-3 or K=3	0.74	0.61	0.44	0.42
1-4 or K=4	0.65	0.52	0.35	0.29

Table 4: PINN #1 and PINN #2 model performance with 30% training dataset for predicting shear stress and slip rate with fine-tuning.

Frozen Layer	Shear Stress R ²		Slip Rate R ²	
	TL: PINN #1	TL: PINN #2	TL: PINN #1	TL: PINN #2
None or K=0	0.84	0.83	0.80	0.78
1 or K=1	0.82	0.72	0.63	0.52
1-2 or K=2	0.81	0.70	0.67	0.60
1-3 or K=3	0.76	0.70	0.59	0.43
1-4 or K=4	0.61	0.53	0.38	0.33

Table 5: PINN #1 and PINN #2 model performance with 10% training dataset for predicting shear stress and slip rate with fine-tuning.

Frozen Layer	Shear Stress R ²		Slip Rate R ²	
	TL: PINN #1	TL: PINN #2	TL: PINN #1	TL: PINN #2
None or K=0	0.79	0.75	0.84	0.8
1 or K=1	0.72	0.61	0.50	0.43
1-2 or K=2	0.66	0.51	0.54	0.52
1-3 or K=3	0.64	0.50	0.64	0.22
1-4 or K=4	0.66	0.46	0.48	0.38

In addition to the fine-tuning described above, we carried out a study in which we built standalone PINN #1 and PINN #2 models on the p5271 dataset (TL dataset). The corresponding performances are shown in Table 6 below for the models using a split of 70%-10%-20% for training, validation, and testing. According to the results, the PINN #2 model performs better than PINN #1 model, much like standalone models for the p5270 dataset.

Table 6: Performance of the standalone PINN #1 and PINN #2 model on the p5271 dataset (TL dataset).

Data	Shear Stress R ²		Slip Rate R ²	
	TL: PINN #1	TL: PINN #2	TL: PINN #1	TL: PINN #2
Training	0.96	0.97	0.86	0.92
Validation	0.92	0.92	0.91	0.91
Testing	0.78	0.80	0.73	0.74

These studies support our observation that in the TL case, the PINN #1 models consistently perform better than PINN #2 models possibly because they are less constrained (exclusion of ultrasonic attributes). We do not have other explanation for this observation. To address this, the following statement is added to the revised manuscript in lines 213 to 215:

“Further model tuning with cosine decay schedule and fine-tuning (freezing one or more layers) show that the TL PINN #1 models consistently outperform the TL PINN #2 models in predicting shear stress and slip rate in all scenarios.”

2. In terms of minor comments, I suggest capitalizing "physics-informed neural network" to emphasize the definition of the acronym PINN, both at line 14 and when it is redefined at line 49. Also, "SHM" is defined as an acronym but only used once; therefore, you may consider removing it at line 17.

Thank you. We modified lines 14 and 49 from “physics-informed neural network” to “Physics-Informed Neural Network” to emphasize the definition of the acronym PINN, and we removed the acronym “SHM”.

3. Regarding Figure 1a, the shear stress and slip rate are encoded with the same color, and the line styles do not allow distinguishing them. Would it be possible to differentiate them with two colors? Additionally, in Figure 1b, you may consider using the label "Normalized shear stress and slip rate" instead of "Normalized", and the "Time" label may also refer to "Ultrasonic signal local time".

We revised Figure 1a according to your suggestion. The slip rate is now plotted in black and shear stress is in purple. The line widths are also increased to make the figure clearer and more readable. In Figure 1b, the left y-axis label is changed from “Normalized” to

“Normalized Shear Stress & Slip Rate” and the right y-axis label is changed from “Time” to “Ultrasonic Signal Local Time”.

Figure 1: Friction experiment coupled with ultrasonic monitoring: schematic setup and typical data. (a) Temporal evolution of shear stress and slip rate in experiment p5270. The inset shows a schematic of the DDS setup with two ultrasonic transducers (transmitter T and receiver R) probing the fault. The thin vertical dashed line corresponds to the time at which the reference ultrasonic waveform is chosen (see text for more details). (b) Schematic representation of active-source ultrasonic monitoring during the experiment. The ultrasonic waveforms are recorded every millisecond throughout the stick-slip cycles. Only a small subset of the waveforms is shown for readability. (c) An example of a recorded ultrasonic signal. Input features to the machine learning models are extracted from the initial portion of the ultrasonic signals (highlighted in brown).

4. In the Ultrasonic Feature Extraction section, could you please clarify how you select the reference waveform, and whether your results are reference-dependent?

The reference waveform S_0 is chosen past the peak friction, and right before the fault starts its transition from stable sliding to unstable seismic cycles as marked in Figure 1a in the revised manuscript.

The two features, namely wave speed v_i and spectral amplitude A_i , are extracted from the recorded waveforms. The amplitude A_i is calculated using the Fourier transform of the recorded signals so it is independent of the reference waveform. The extraction of v_i involves the time of flight TOF_i calculations that consider the arrival time TOF_0 (reference

waveform) and estimated time delay Δt_i , which is calculated through cross-correlation. Since the waveform shapes change only slightly and cross-correlation coefficients for the recorded waveforms always remain higher than 0.97, the choice of reference waveform does not significantly affect the extracted feature v_i . While we have not conducted a systematic study to investigate this (i.e., using datasets with different reference waveforms as input to the ML models), it is our experience that when the correlation coefficient is very close to one, the extracted wave speeds are unaffected by the choice of reference.

To address this point, lines 115 to 119 in the original manuscript are rewritten as follows:

“To calculate the evolution of wave speed during frictional sliding, we first extract the time delay Δt by cross-correlating each waveform S_i with a reference waveform S_0 . The reference waveform is chosen past the peak friction just before the fault starts its transition from stable sliding to unstable seismic cycles (thin vertical dashed line at time = 2065 s in Figure 1a). The shape of the recorded waveforms S_i changes little throughout the experiment such that the cross-correlation coefficient remains always greater than 0.97.”

5. Finally, PZT is defined but used only once at line 102.

Thank you for pointing this out. The acronym PZT was removed from line 102.

Reviewer #3 (Remarks to the Author):

Comments for Borate et al. "Using a physics-informed neural network and fault zone acoustic monitoring to predict lab earthquakes "

This study shows the superiority of physics-informed neural networks (PINN) compared to data-driven neural networks for predicting shear stress history and slip rates in laboratory shear-slip experiments. The structure of the paper is straightforward and clearly demonstrates the superiority of PINN, and I did not find any particular problem. The manuscript is clearly written, and I personally think it is almost ready for publication. One concern is that the comparison target is a simple multilayer perceptron neural network that is not expected to perform well, so the paper is not very interesting in terms of performance. However, this study shows that PINN, which includes simple physical rules, can be effective to a certain extent, and I consider this study worth publishing.

1. P.3 L 104: Are the piezoelectric transducers used as a transmitter and a receiver the same type? If possible, it would be desirable to specify the manufacturer and model number.

Yes, the same type of piezoelectric transducer is used for the transmitter and the receiver. The piezoelectric disks are 12.7 mm in diameter, 4 mm thick made of material 850 from American Piezo Ceramics (APC International). This information is now included in the revised manuscript in lines 104 to 106 as follows:

"The two identical piezoelectric disks, used as transmitter and receiver, are 12.7 mm in diameter, 4 mm thick (corresponding to a center frequency of 500 kHz) made of material 850 from American Piezo Ceramics (APC International)."

2. P. 4 L 123: Does this 10-point backward-looking average affect the results?

Yes, the feature smoothing is part of data pre-processing and it does improve the data-driven model performance (Shreedharan et al., 2021). We note that the averaging has been done carefully (backward-looking) and the same pre-processed data are used for the data-driven and PINN models.

3. P. 10 L 301 I did not understand how the parameters of σ , K , ... were obtained in the PINN approach. Could you add the description?

The actual values of some experimental parameters are known (normal stress (σ), shearing rate (V_l), and density (ρ)) or measurable (system stiffness (k)). But to avoid any errors due to the unit mismatch between the features (spectral amplitude, wave speed), outputs (shear stress, slip rate), and these constants in the physics constraints, we treat them as learnable parameters that are learned alongside neural network parameters such as weights and biases in the proposed PINN framework. In addition, this provides an opportunity to check the model's working by comparing the learned vs true parameters.

These weights are updated during model training using stochastic gradient descent and its variants. Once the model is fully trained, these learned weights are extracted from the model layers, scaled back (to undo normalization), and compared with the true available experimental values as shown in Table 2 of the originally submitted manuscript. This gives us the chance to look into the models' ability to correctly learn the values of these constants.

To address this comment following paragraph is added in the revised manuscript in lines 325 to 334:

Although these parameters are either known (σ , V_I , ρ , k) or measurable (A_{Intact}) could be measured by testing an intact granite block of the same thickness as the cumulative thickness of the blocks used in the friction experiment), we treat them as trainable neural network weights in the PINN framework. This approach is used to avoid errors due to unit mismatch between features, outputs, and these constants in the constraints.

These weights are extracted from the layers of the fully trained models and converted back to the original scale to undo the effect of data normalization (see implementation details in the https://github.com/prabhavorate92/PINN_Paper.git). A comparison of the scaled learned weights with the known parameter values gives us the opportunity to examine the PINN model by determining how well the models are able to learn the values of parameters measured experimentally.

4. Fig. 1 It is difficult to distinguish shear stress and slip rate in this figure. I want to suggest using different colors.

Thank you. Reviewer 2 made the same comment. We have revised Figure 1 as shown below. The slip rate is now plotted in black and shear stress is in purple. The line widths are also increased to make the figure clearer and more readable.

Figure 1: Friction experiment coupled with ultrasonic monitoring: schematic setup and typical data. (a) Temporal evolution of shear stress and slip rate in experiment p5270. The inset shows a schematic of the DDS setup with two ultrasonic transducers (transmitter T and receiver R) probing the fault. The thin vertical dashed line corresponds to the time at which the reference ultrasonic waveform is chosen (see text for more details). (b) Schematic representation of active-source ultrasonic monitoring during the experiment. The ultrasonic waveforms are recorded every millisecond throughout the stick-slip cycles. Only a small subset of the waveforms is shown for readability. (c) An example of a recorded ultrasonic signal. Input features to the machine learning models are extracted from the initial portion of the ultrasonic signals (highlighted in brown).

5. The authors used a simple perceptron model as a data-driven model. In my opinion, for such time series data, the convolutional neural network approach can obtain better performance, possibly comparable to PINN results. Are the authors performed such validation?

Thank you for this comment. We did perform additional results using the convolution layer to test whether the filters can stably extract the local temporal information from the data. However, as discussed in this work, physics constraints reduces the overall parameter search space and help the model generalize without compromising the

convergence speed. Complex models such as CNN due to tensor weights often lead to over-parameterization leading to sub-optimal results while working with temporal dynamics signals, the ones used in this study. Our results validate this hypothesis. In our experiments CNN has the same set of parameters as our MLP and PINN, we did this to ensure the memorization effect due to higher parameter count can be avoided and we can have a fair comparison with our models.

Here, we discuss the results obtained when implementing a convolutional neural network (CNN) on the p5270 dataset used in this study. The CNN architecture consists of a standard setup comprising of convolutional, activation, max pooling, and flatten layers as shown in Figure 1 below. Similar to our MLP models, the two input features (wave amplitude and speed) are provided with 3-sec history (300 samples for each feature, i.e., 600 samples in total) before the current time to predict shear stress and slip rate at the current time. The CNN layers have tensor weights, that convolve across features to generate the weighted sum and are then passed through the ReLU activation function, we then apply max-pooling over the output of the last layer. The generated 2D arrays from pooled feature maps are all flattened into a single, lengthy continuous linear vector. To estimate shear stress and slip rate, this vector is provided as input to the fully connected layer. The CNN model is implemented with a Training-Validation-Testing split of 70%-10%-20%, and optimized using mean squared error. Furthermore, we perform a grid search over learning rate and batch size to find the optimal hyperparameter for our CNN model. The performance of the model (R^2 score) in predicting shear stress and slip rate is tabulated in Tables 1 and 2 respectively. Although CNN is known to be effective at extracting discrete signal features, the poor model performance suggests that it is not responsive to temporal dynamics, which is essential here.

Furthermore, we have done another study on the same p5270 dataset (manuscript in preparation), which is relevant to this discussion. In this study, we use CNN layers to (automatically) extract features from raw ultrasonic signals recorded during stick-slip experiments. The best shear stress prediction performance using these automatically extracted features for a model developed with a training-validation-test split of 70%-10%-20% is provided in Table 3. A comparison with the results presented in this paper indicates that CNN-extracted features do not lead to a better performance than hand-picked features (wave speed, wave amplitude, and center frequency).

These studies show that CNN models do not perform better than MLP models. Although using CNN layers is appropriate for extracting features from raw ultrasonic signals, due to the averaging involved, the resulting models are inferior to the models that use expert-crafted features (wave speed, amplitude, and center frequency).

Figure 1: CNN model for shear stress and slip rate prediction.

Table 1: Shear stress prediction R^2 scores for training (70%)-validation (10%)-testing (20%) dataset.

Model	Training R^2	Validation R^2	Testing R^2
Data-driven	-228.02	-155.04	-160.82
PINN 1	-85.84	-58.79	-59.88
PINN 2	-14.17	-11.99	-12.95

Table 2: Slip rate prediction R^2 scores for training (70%)-validation (10%)-testing (20%) dataset.

Model	Training R^2	Validation R^2	Testing R^2
Data-driven	-4.80	-3.25	-3.51
PINN 1	-3550.88	-2198.01	-2343.76
PINN 2	-125.17	-79.12	-84.06

Table 3: Shear stress prediction performance from automatically extracted features using a CNN framework.

Parameter	Training R^2	Validation R^2	Testing R^2
Shear Stress	0.69	0.77	0.58

References

- Avouac, J. P. (2015). From geodetic imaging of seismic and aseismic fault slip to dynamic modeling of the seismic cycle. *Annual Review of Earth and Planetary Sciences*, 43, 233–271. <https://doi.org/10.1146/annurev-earth-060614-105302>
- Barbot, S., Lapusta, N., & Avouac, J. P. (2012). Under the hood of the earthquake machine: Toward predictive modeling of the seismic cycle. *Science*, 336(6082), 707–710. <https://doi.org/10.1126/science.1218796>
- Blanke, A., Kwiatek, G., Goebel, T. H. W., Bohnhoff, M., & Dresen, G. (2021). Stress drop-magnitude dependence of acoustic emissions during laboratory stick-slip. *Geophysical Journal International*, 224(2), 1371–1380. <https://doi.org/10.1093/gji/ggaa524>
- Bolton, D. C., Marone, C., Shokouhi, P., Rivière, J., Rouet-Leduc, B., Hulbert, C., & Johnson, P. A. (2019). Characterizing acoustic signals and searching for precursors during the laboratory seismic cycle using unsupervised machine learning. *Seismological Research Letters*, 90(3), 1088–1098. <https://doi.org/10.1785/0220180367>
- Bolton, D. C., Shreedharan, S., Rivière, J., & Marone, C. (2020). Acoustic Energy Release During the Laboratory Seismic Cycle: Insights on Laboratory Earthquake Precursors and Prediction. *Journal of Geophysical Research: Solid Earth*, 125(8). <https://doi.org/10.1029/2019JB018975>
- Bu, F., Xue, L., Zhai, M., Huang, X., Dong, J., Liang, N., & Xu, C. (2022). Evaluation of the characterization of acoustic emission of brittle rocks from the experiment to numerical simulation. *Scientific Reports*, 12(1), 1–16. <https://doi.org/10.1038/s41598-021-03910-8>
- Bürgmann, R., Ayhan, M. E., Fielding, E. J., Wright, T. J., McClusky, S., Aktug, B., Demir, C., Lenk, O., & Türkezer, A. (2002). Deformation during the 12 November 1999 Düzce, Turkey, earthquake, from GPS and InSAR data. *Bulletin of the Seismological Society of America*, 92(1), 161–171. <https://doi.org/10.1785/0120000834>
- Chow, T. M., Meglis, I. L., & Young, R. P. (1995). Progressive microcrack development in tests on Lac du Bonnet granite-II. Ultrasonic tomographic imaging. *International Journal of Rock Mechanics and Mining Sciences And*, 32(8), 751–761. [https://doi.org/10.1016/0148-9062\(95\)00015-9](https://doi.org/10.1016/0148-9062(95)00015-9)
- Dunegan, H., & Harris, D. (1969). Acoustic emission—a new nondestructive testing tool. *Ultrasonics*, 7(3), 160–166. [https://doi.org/10.1016/0041-624X\(69\)90660-X](https://doi.org/10.1016/0041-624X(69)90660-X)
- Fielding, E. J., Sladen, A., Li, Z., Avouac, J. P., Bürgmann, R., & Ryder, I. (2013). Kinematic fault slip evolution source models of the 2008 M7.9 wenchuan earthquake in china from SAR interferometry, GPS and teleseismic analysis and implications for longmen shan tectonics. *Geophysical Journal International*, 194(2), 1138–1166. <https://doi.org/10.1093/gji/ggt155>
- Goebel, T. H. W., Becker, T. W., Schorlemmer, D., Stanchits, S., Sammis, C., Rybacki, E., & Dresen, G. (2012). Identifying fault heterogeneity through mapping spatial anomalies in acoustic emission statistics. *Journal of Geophysical Research: Solid Earth*, 117(3), 1–18.

<https://doi.org/10.1029/2011JB008763>

- Goebel, T. H. W., Schorlemmer, D., Becker, T. W., Dresen, G., & Sammis, C. G. (2013). Acoustic emissions document stress changes over many seismic cycles in stick-slip experiments. *Geophysical Research Letters*, *40*(10), 2049–2054. <https://doi.org/10.1002/grl.50507>
- Gualandi, A., Perfettini, H., Radiguet, M., Cotte, N., & Kostoglodov, V. (2017). GPS deformation related to the Mw 7.3, 2014, Papanao earthquake (Mexico) reveals the aseismic behavior of the Guerrero seismic gap. *Geophysical Research Letters*, *44*(12), 6039–6047. <https://doi.org/10.1002/2017GL072913>
- Hearn, E. H., & Bürgmann, R. (2005). The effect of elastic layering on inversions of GPS data for coseismic slip and resulting stress changes: Strike-slip earthquakes. *Bulletin of the Seismological Society of America*, *95*(5), 1637–1653. <https://doi.org/10.1785/0120040158>
- Hulbert, C., Rouet-Leduc, B., Johnson, P. A., Ren, C. X., Rivière, J., Bolton, D. C., & Marone, C. (2019). Similarity of fast and slow earthquakes illuminated by machine learning. *Nature Geoscience*, *12*(1), 69–74. <https://doi.org/10.1038/s41561-018-0272-8>
- Jansen, D. P., Carlson, S. R., Young, R. P., & Hutchins, D. A. (1993). Ultrasonic imaging and acoustic emission monitoring of thermally induced microcracks in Lac du Bonnet granite. *Journal of Geophysical Research*, *98*(B12). <https://doi.org/10.1029/93jb01816>
- Jasperon, H., Bolton, D. C., Johnson, P., Guyer, R., Marone, C., & de Hoop, M. V. (2021). Attention Network Forecasts Time-to-Failure in Laboratory Shear Experiments. *Journal of Geophysical Research: Solid Earth*, *126*(11), 1–20. <https://doi.org/10.1029/2021JB022195>
- Kano, M., Fukuda, J., Miyazaki, S., & Nakamura, M. (2018). Spatiotemporal Evolution of Recurrent Slow Slip Events Along the Southern Ryukyu Subduction Zone, Japan, From 2010 to 2013. *Journal of Geophysical Research: Solid Earth*, *123*(8), 7090–7107. <https://doi.org/10.1029/2018JB016072>
- Laurenti, L., Tinti, E., Galasso, F., Franco, L., & Marone, C. (2022). Deep learning for laboratory earthquake prediction and autoregressive forecasting of fault zone stress. *Earth and Planetary Science Letters*, *598*, 117825. <https://doi.org/10.1016/j.epsl.2022.117825>
- Lockner, D. (1993). The role of acoustic emission in the study of rock fracture. *International Journal of Rock Mechanics and Mining Sciences And*, *30*(7), 883–899. [https://doi.org/10.1016/0148-9062\(93\)90041-B](https://doi.org/10.1016/0148-9062(93)90041-B)
- Lubbers, N., Bolton, D. C., Mohd-Yusof, J., Marone, C., Barros, K., & Johnson, P. A. (2018). Earthquake Catalog-Based Machine Learning Identification of Laboratory Fault States and the Effects of Magnitude of Completeness. *Geophysical Research Letters*, *45*(24), 13,269–13,276. <https://doi.org/10.1029/2018GL079712>
- Michel, S., Gualandi, A., & Avouac, J. P. (2019). Similar scaling laws for earthquakes and Cascadia slow-slip events. *Nature*, *574*(7779), 522–526. <https://doi.org/10.1038/s41586-019-1673-6>
- Niu, F., Silver, P. G., Daley, T. M., Cheng, X., & Majer, E. L. (2008). Preseismic velocity changes

- observed from active source monitoring at the Parkfield SAFOD drill site. *Nature*, 454(7201), 204–208. <https://doi.org/10.1038/nature07111>
- Pu, Y., Chen, J., & Apel, D. B. (2021). Deep and confident prediction for a laboratory earthquake. *Neural Computing and Applications*, 33(18), 11691–11701. <https://doi.org/10.1007/s00521-021-05872-4>
- Rivière, J., Lv, Z., Johnson, P. A., & Marone, C. (2018). Evolution of b-value during the seismic cycle: Insights from laboratory experiments on simulated faults. *Earth and Planetary Science Letters*, 482, 407–413. <https://doi.org/10.1016/j.epsl.2017.11.036>
- Rouet-Leduc, B., Hulbert, C., Bolton, D. C., Ren, C. X., Riviere, J., Marone, C., Guyer, R. A., & Johnson, P. A. (2018). Estimating Fault Friction From Seismic Signals in the Laboratory. *Geophysical Research Letters*, 45(3), 1321–1329. <https://doi.org/10.1002/2017GL076708>
- Rouet-Leduc, B., Hulbert, C., Lubbers, N., Barros, K., Humphreys, C. J., & Johnson, P. A. (2017). Machine Learning Predicts Laboratory Earthquakes. *Geophysical Research Letters*, 44(18), 9276–9282. <https://doi.org/10.1002/2017GL074677>
- Savage, J. C., & Hasegawa, H. S. (1964). Some properties of tensile fractures inferred from elastic wave radiation. *Journal of Geophysical Research*, 69(10), 2091–2100. <https://doi.org/10.1029/jz069i010p02091>
- Scholz, C. H. (1968). Microfracturing and the inelastic deformation of rock in compression. *Journal of Geophysical Research*, 73(4), 1417–1432. <https://doi.org/10.1029/jb073i004p01417>
- Shreedharan, S., Bolton, D. C., Rivière, J., & Marone, C. (2021). Machine Learning Predicts the Timing and Shear Stress Evolution of Lab Earthquakes Using Active Seismic Monitoring of Fault Zone Processes. *Journal of Geophysical Research: Solid Earth*, 126(7), 1–18. <https://doi.org/10.1029/2020JB021588>
- Wallace, L. M., Webb, S. C., Ito, Y., Mochizuki, K., Hino, R., Henrys, S., Schwartz, S. Y., & Sheehan, A. F. (2016). Slow slip near the trench at the Hikurangi subduction zone, New Zealand. *Science*, 352(6286), 701–704. <https://doi.org/10.1126/science.aaf2349>
- Wang, K., Johnson, C. W., Bennett, K. C., & Johnson, P. A. (2021). Predicting fault slip via transfer learning. *Nature Communications*, 12(1), 1–11. <https://doi.org/10.1038/s41467-021-27553-5>
- Wang, K., Johnson, C. W., Bennett, K. C., & Johnson, P. A. (2022). Predicting Future Laboratory Fault Friction Through Deep Learning Transformer Models. *Geophysical Research Letters*, 49(19), 1–9. <https://doi.org/10.1029/2022GL098233>

REVIEWERS' COMMENTS

Reviewer #1 (Remarks to the Author):

The authors took all my comments into account and, in my opinion, also addressed the other reviewers' comments well. I recommend the manuscript for publication with no further changes.

Reviewer #2 (Remarks to the Author):

Dear Authors and Esteemed Editor,

I have read the revised version of the manuscript NCOMMS-22-50565A entitled "Using a Physics-Informed Neural Network and Fault Zone Acoustic Monitoring to Predict Lab Earthquakes" proposed by Parisa Shokouhi and collaborators for Nature Communications.

In this new version, the authors carefully addressed all the points I raised. Their answers and corrections are correct to me and have helped clarify the manuscript. I am in favor of accepting the manuscript as-is.

Yours sincerely.